# Synthesis of Soluble High Molar Mass Poly(Phenylene Methylene)-Based Polymers

**DOI:** 10.3390/polym16070967

**Published:** 2024-04-02

**Authors:** Marco F. D’Elia, Yingying Yu, Melvin Renggli, Madeleine A. Ehweiner, Carina Vidovic, Nadia C. Mösch-Zanetti, Markus Niederberger, Walter Caseri

**Affiliations:** 1Laboratory for Multifunctional Materials, Department of Materials, ETH Zürich, 8093 Zürich, Switzerland; yingyu@student.ethz.ch (Y.Y.); rengglme@student.ethz.ch (M.R.); markus.niederberger@mat.ethz.ch (M.N.); 2Institut für Chemie/Bereich Anorganische Chemie, Universitaet Graz, Schubertstraße 1/3, 8010 Graz, Austria; madeleine.ehweiner@uni-graz.at (M.A.E.); vidovic.carina@web.de (C.V.); nadia.moesch@uni-graz.at (N.C.M.-Z.)

**Keywords:** poly(phenylene methylene), catalysts, high molar mass, fractionation

## Abstract

Poly(phenylene methylene) (PPM) is a multifunctional polymer that is also active as an anticorrosion fluorescent coating material. Although this polymer was synthesized already more than 100 years ago, a versatile synthetic route to obtain soluble high molar mass polymers based on PPM has yet to be achieved. In this article, the influence of bifunctional bis-chloromethyl durene (BCMD) as a branching agent in the synthesis of PPM is reported. The progress of the reaction was followed by gel permeation chromatography (GPC) and NMR analysis. PPM-based copolymers with the highest molar mass reported so far for this class of materials (up to *M*_n_ of 205,300 g mol^−1^) were isolated. The versatile approach of using BCMD was confirmed by employing different catalysts. Interestingly, thermal and optical characterization established that the branching process does not affect the thermoplastic behavior and the fluorescence of the material, thus opening up PPM-based compounds with high molar mass for applications.

## 1. Introduction

Poly(phenylene methylene) (PPM) (Figure 1a), a polymer consisting of phenylene units linked by methylene bridges [1,2,3], exhibits high thermal stability [4,5,6], hydrophobicity [5] and fluorescence [7,8,9]. Accordingly, PPM was proposed for a wide range of potential applications such as organic coatings for corrosion prevention [10,11], adhesives [10] and aerospace applications [2,10]. The first synthesis of PPM dates back to Cannizzaro in 1853 [12], and since then, poly(phenylene methylene) was usually obtained by polymerizing benzyl chloride or benzyl alcohol with Lewis acids such as SnCl_4_, FeCl_3_ or AlCl_3_ [13,14,15,16,17] as catalysts. However, the molar masses of isolated PPM products have been low (number average molar mass, *M*_n_, commonly below 6500 g mol^−1^) [3,16,18] and this was also considered to be the reason for the high brittleness of this kind of polymer [19,20]. Recent optimization of the reaction conditions of the bulk polymerization of benzyl chloride under the action of SnCl_4_ and FeCl_3_ yielded isolated PPM with *M*_n_ ranging from 8000 g mol^−1^ to 10,000 g mol^−1^ [21]. Other catalysts were also employed for the polymerization of benzyl chloride, e.g., compounds of W, Mo or Cr which yielded low molar mass PPM [10,22,23]. Recently, dibromotungsten(II) complexes were found to be effective in the polymerization of benzyl chloride, resulting in poly(phenylene methylene)s of *M*_n_ in the range of 8000 g mol^−1^ to 15,000 g mol^−1^ [24]. These products showed a multimodal and thus an extremely broad molar mass distribution (polydispersity index (PDI) between 16.7 and 33.9), and, by fractionation, a product with *M*_n_ of 167,900 g mol^−1^ was isolated [24]. This is, to our best knowledge, the highest *M*_n_ determined so far for a poly(phenylene methylene).

Alternatively, copolymerization with bifunctional comonomers such as dichloroxylene (ClCH_2_C_6_H_4_CH_2_Cl) also led to an increase in molar mass, although to such an extent that highly crosslinked thermosetting products were obtained [25,26,27]. Thus, the polymers obtained in this way were insoluble and essentially non-processable. Crosslinking is probably enhanced by the phenylene units of the xylene moiety which allow additional branching by substitution of the hydrogen atoms of those phenylene units [25,26,27]. Therefore, extensive crosslinking should be avoided by the use of tetramethyl-substituted dichloroxylene, i.e., 1,4-bis(chloromethyl)-2,3,5,6-tetramethylbenzene (3,6-bis(chloromethyl)durene, BCMD, Figure 1b. In order to show the potential of BCMD in increasing *M*_n_ and the possibility of isolating PPM of high molar masses, not only was a common Lewis acid employed as a catalyst, as already reported for PPM-based polymers [28], but new catalysts based on tungsten and molybdenum complexes were also evaluated for the polymerization of benzyl chloride, and one of them was subsequently selected for copolymerization with BCMD. We also followed the molar mass evolution during polymerization as a function of monomer conversion in order to adjust the reaction parameters. 

## 2. Materials and Methods

### 2.1. Materials

Benzyl chloride (99%, containing propylene oxide as stabilizer), methanol, chloroform, tetrahydrofuran (THF) (≥99.9% HPLC LiChrosol), 2-butanone and toluene were all purchased from Sigma-Aldrich (Buchs, Switzerland). The catalysts [WCl_4_(THF)_2_], [WCl_4_(MeCN)_2_], [WBr_2_(CO)_3_(dme)] and [MoI_2_(CO)_3_(MeCN)_2_] used in this article and displayed in Figure 2 were stored under inert gas atmosphere before use (for synthesis, see Appendix A). 

### 2.2. Synthesis of Poly(Phenylene Methylene) Catalyzed by Complexes Based on W and Mo

The polymerizations of benzyl chloride were carried out with a monomer-to-catalyst ratio of about 0.1% mol/mol for each catalyst. Although the polymerization conditions were essentially based on procedures already reported in the literature [9,11,21], the reaction temperature had to be modified for the catalytic systems applied here, independently of time constraints, from room temperature to 80 °C, 120 °C, 160 °C and 180 °C in order to mitigate the viscosity increase and allow an efficient mixing over the course of the reaction. Below, the example of the polymerization catalysed by [WCl_4_(THF)_2_] is provided. The stabilizer propylene oxide present in the starting material was removed from benzyl chloride under reduced pressure (ca. 10^−2^ bar) overnight, as already reported before [9,11,21] (the success of the procedure was controlled by ^1^H NMR spectroscopy). In a 50 mL three neck flask, 20 g of benzyl chloride (20.8 mL, 0.16 mol) was added to the solid catalyst [WCl_4_(THF)_2_] (70 mg, 0.1 mmol) under nitrogen atmosphere keeping a constant gas flow of 15 mL min^−1^. The initial reaction mixture was then mechanically stirred for 3 h in order to ensure a good mixing between the catalyst and the monomer. Over the course of reaction, the temperature was increased from 25 °C to 180 °C in order to enable mixing upon the increase in viscosity due to the molar mass increase. The heating conditions are shown in Figure 3, and the monomer conversion was constantly evaluated by sampling aliquots from the reaction mixture. As the reaction was complete, the molten polymer was allowed to cool down to room temperature. The product was purified by dissolving the polymer in 30 mL of chloroform and then pouring the solution into 600 mL of methanol. The suspension was vigorously stirred for 3 h. The obtained PPM powder was filtered over cellulose filter and the polymer powder was dried under vacuum (≈10^−2^ bar) overnight. A quantity of 6.3 g of pale-yellow polymer was obtained (yield 66%). ^1^H NMR (300 MHz, CDCl_3_, δ in ppm): 3.79 (broad, 2H), 7 (broad, 4H). Molar masses obtained by GPC analysis are given in Table 1 and the ^13^C NMR spectrum is shown below.

The synthesis with the catalysts [WCl_4_(MeCN)_2_], [WBr_2_(CO)_3_(dme)], [MoI_2_(CO)_3_(MeCN)_2_] was performed analogously and the respective temperature ramps are reported Results and Discussions. The yields of purified PPM polymers amounted to 66%–77% and are indicated in Results and Discussions section together with the molar masses. The ^13^C NMR and ^1^H NMR spectra are presented in the Appendix A.

### 2.3. Synthesis of PPM with Durene Units (PPM-D)

PPM with durene units was synthesized in the presence of 0.5% mol/mol 1,4-bis(chloromethyl)-2,3,5,6-tetramethylbenzene (3.6-bis(chloromethyl)durene, BCMD) as described above with [W_2_Cl_4_(THF)_2_] (75 mg, 0.16 mmol) as the catalyst; however, in this case we added 172 mg of BCMD (7.4·10^−1^ mmol) to 17 mL benzyl chloride (148 mmol). The evolution of color during the reaction was as follows: clear yellow brown for the first minute, black at 80 °C, blue at 120 °C, and dark green at 160 °C. After sample work-up as described above for PPM, a quantity of 6.63 g (82%) of green bluish product was obtained. ^1^H NMR (300 MHz, CDCl_3_, δ in ppm): 2.5 (s, 0.12H, CH_3_) 3.71 (br, 2H, CH_2_), 7.19 (br, 4H, Ar) (spectra shown in Appendix A); GPC (CHCl_3_): *M*_n_ = 3400 g mol^−1^, weight average molar mass (*M*_w_) = 211,977 g mol^−1^, *M*_w_/*M*_n_ = 55.4; DSC (*T*_g_): 52.0 °C.

### 2.4. Fractionation

The copolymer (1 g) (*M*_n_ = 3317 g mol^−1^, *M*_w_ = 183,600 g mol^−1^) and 2-butanone (23 mL) were stirred vigorously for 2 h after which the suspension separated into a clear upper phase with the low molar mass polymer (F_low_) and a turbid oily phase with the high molar mass polymer (F_medium_). The upper and the lower phases were separated, and the solvent was removed by a rotary evaporator, and the remaining products were dissolved again in 5 mL of chloroform. The solutions were poured in 200 mL of methanol under stirring, and the precipitated solids were filtered and dried (as described above), to give 0.452 g (fractionation yield 45%) of F_medium_ (*M*_n_ = 33,520 g mol^−1^, *M*_w_ = 322,000 g mol^−1^) (when the fractionation procedure was repeated twice, it was not possible to isolate higher molar mass fractions). Thereafter, 50 mg of F_medium_ were further washed with 5 mL of a mixture of chloroform/2-butanone (1:1 by volume) in order to remove lower molar mass fractions and to provide the polymer fraction F_high_ (23 mg, fractionation yield 46%) (*M*_n_ = 205,300 g mol^−1^, *M*_w_ = 777,900 g mol^−1^).

### 2.5. Characterization

The ^1^H NMR and ^13^C NMR spectra were recorded on a Bruker AV300 MHz spectrometer (Billerica, MA, USA) using CDCl_3_ as the solvent. The multiplicity of peaks is indicated as (bs) for broad signals, (s) singlet, (d) doublet, (t) triplet and (m) multiplet. The monomer conversion χ was evaluated by withdrawing aliquots of the reaction mixtures during the reaction to be analyzed by ^1^H NMR spectroscopy according to the literature [21,24].

The molar masses were investigated by gel permeation chromatography (GPC) using a Viscotek GPC system using tetrahydrofuran (THF) as eluent. The GPC module comprised a pump and degasser system (GPCmax VE2001, Malvern, Worcs, UK; 1.0 mL min^−1^ flow rate), a Viscotek 302 TDA unit as detector (Malvern, Worcs, UK) and two columns for the analysis of different molar masses (2× PLGel Mix-B; dimensions 7.5 mm × 300 mm (all supplied by Malvern, Worcs, UK). The thermal characterization was performed with a TGA/DSC 3+ module (Mettler Toledo, Schwerzenbach, Switzerland). The thermal transitions were investigated from 25 °C to 360 °C under nitrogen flush (50 mL min^−1^), increasing the temperature with a rate of 10 °C min^−1^. The onset of decomposition was evaluated in a temperature range of 25 °C to 900 °C under air flush (50 mL min^−1^) with a temperature increasing rate of 10 °C min^−1^.

## 3. Results and Discussions

### 3.1. Homopolymerization of Benzyl Chloride with Different Catalysts

The screening of [WCl_4_(MeCN)_2_], [WCl_4_(THF)_2_], [WBr_2_(CO)_3_(dme)] and [MoI_2_(CO)_3_(MeCN)_2_] as catalysts for the bulk polymerization of benzyl chloride was performed keeping the same molar ratio catalyst/monomer (0.1% mol/mol) for all the compounds. Before the temperature increase and the start of the reactions, the catalysts were dissolved in benzyl chloride at room temperature. Due to the different solubilities of each catalyst in benzyl chloride, dissolution times in the range of minutes were observed for [MoI_2_(CO)_3_(MeCN)_2_] and [WBr_2_(CO)_3_(dme)], while for the W(IV)-based catalysts, a time of 4 h was needed. The temperature of the reaction was then adjusted over the course of polymerization to avoid mixing problems due to the increase in viscosity (i.e., the Weissenberg effect). The temperature required for polymerization and the consequent monomer conversion strongly depended on the compound as displayed in Figure 3a,b. In particular, the polymerization in the presence of [MoI_2_(CO)_3_(MeCN)_2_] was triggered already at 80 °C, quickly reaching (10 min) a monomer conversion of about 90%. When [WCl_4_(MeCN)_2_] was employed, the monomer conversion raised significantly between 80 °C to 120 °C settling above 80% after 5 h at this temperature. Previously reported W(II)-based catalysts also showed catalytic activity below 120 °C [24]. By contrast, polymerizations catalyzed by [WCl_4_(THF)_2_] or [WBr_2_(CO)_3_(dme)] were initiated only at temperatures at or above 150 °C.

**Figure 3 polymers-16-00967-f003:**
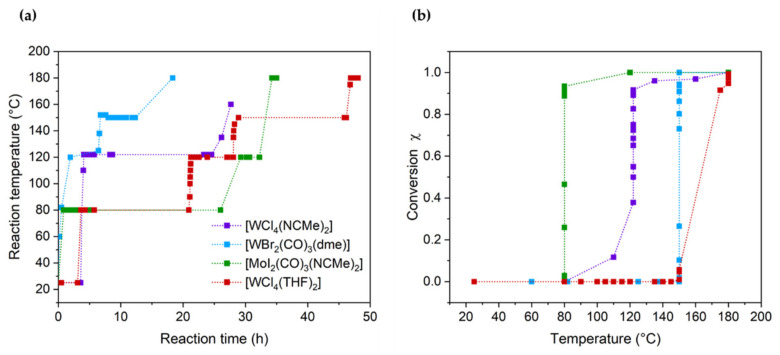
(**a**) Temperature ramp for each reaction catalyzed by [WCl_4_(MeCN)_2_] (violet), [WCl_4_(THF)_2_] (red), [WBr_2_(CO)_3_(dme)] (blue) and [MoI_2_(CO)_3_(MeCN)_2_] (green) during the heating process. (**b**) Monomer conversion during the polymerization reactions with each catalyst as a function of temperature.

The monomer conversion of the polymerization catalyzed by [WBr_2_(CO)_3_(dme)] reached 100% after 5 h at 150 °C. Notably, at this temperature, no striking increase in viscosity was observed, probably due to the low molar mass of the obtained polymer. On the other hand, the monomer conversion at 150 °C of the polymerization catalyzed by [WCl_4_(THF)_2_] settled below 10% after 17 h. Hence, the rise in the temperature to 180 °C was crucial in order to increase the monomer conversion and complete the reaction as evident from Figure 4a, which shows the monomer conversion obtained from the ^1^H NMR spectra of aliquots removed from the reaction mixture. Figure 4b displays the GPC chromatograms of the aliquots sampled after various monomer conversions. Those chromatograms disclose that upon the triggering of the reaction at 150 °C at the monomer conversion of 2%, a bimodal molar mass distribution with a peak at a retention time of 15.4 min corresponding to a molar mass between 4500 g mol^−1^ and 63,000 g mol^−1^ and a smaller peak at 16.6 min (molar mass below 4480 g mol^−1^) emerged. After 17 h at this temperature, the two peaks shifted to lower retention times, 14.3 min and 15.4 min, both in the molar mass range between 4000 g mol^−1^ and 500,000 g mol^−1^. Those chromatograms showed a tail after 17 min corresponding to monomers and oligomers still present in the reaction batch at this degree of monomer conversion. As the temperature was increased up to 180 °C, the oligomer fractions grew faster with respect to the high molar mass fraction reflecting a rise in the peak at 16.3 min and thus a pronounced decrease in *M*_n_ as the monomer conversion increased (Figure 4a). The opposite was observed for other reported tungsten-based catalysts [24].

The presence of high molar masses at low monomer conversions indicates that in the presence of [WCl_4_(THF)_2_], a chain-growth-like mechanism is involved (a chain-growth-like process was also reported for other tungsten-based catalysts [24]). By contrast, such an effect was not observed in the polymerizations using [WCl_4_(MeCN)_2_], [WBr_2_(CO)_3_(dme)] and [MoI_2_(CO)_3_(MeCN)_2_].

After isolation of the polymers by dissolution and subsequent precipitation, the polymers resulting from the [WCl_4_(MeCN)_2_], [WBr_2_(CO)_3_(dme)] and [MoI_2_(CO)_3_(MeCN)_2_] catalysts (Figure 5) exhibited a monomodal molar mass distribution with *M*_n_ between 3100 g mol^−1^ and 4500 g mol^−1^ and *M*_w_ between 7000 g mol^−1^ and 13,000 g mol^−1^ (Table 1), which lies in the conventional range of PPM (see Section 1, Introduction). However, the PPM obtained using [WCl_4_(THF)_2_] revealed a bimodal molar mass distribution (Figure 5), presenting a *M*_n_ of 4090 g mol^−1^ in coherence with the other catalysts, but a higher *M*_w_ (63,760 g mol^−1^) and thus a much higher PDI (15.6) (Table 1).

**Table 1 polymers-16-00967-t001:** Number average molar mass (*M*_n_), weight average molar mass (*M*_w_), polydispersity index (PDI) and yield of reaction of the polymers synthesized with the W- and Mo-based catalysts.

Catalyst	*M*_n_ (g mol^−1^)	*M*_w_ (g mol^−1^)	PDI	Yield of Reaction
[WCl_4_(MeCN)_2_]	3118	7013	2.2	68%
[WBr_2_(CO)_3_(dme)]	3325	13,170	3.9	72%
[MoI_2_(CO)_3_(MeCN)_2_]	4538	11,480	2.5	77%
[WCl_4_(THF)_2_]	4090	63,760	15.6	69%

Among the investigated catalysts, the Mo(II)-based complex and [WCl_4_(THF)_2_] led to a *M*_n_ which was 25–30% above that of the other tungsten catalysts. However, we consider this and the higher *M*_w_ obtained with [WCl_4_(THF)_2_] as a specific effect of the applied compounds and not as a general property of molybdenum- or W(IV)-based catalysts, all the more as the catalysts have different operation temperatures.

The ^13^C NMR spectra of the purified polymers (Figure 6) correspond to those of PPM reported with other catalysts [9,21]. The signals in the range of 33 ppm–44 ppm (Figure 6) are attributed to the substitution patterns along the PPM backbone, as reported previously [22] (the ^13^C NMR signals in the aromatic region between 125 ppm and 145 ppm are shown in the Appendix A). The ^1^H NMR spectra (Appendix A) show the broad peak at 3.7 ppm in the methylene region and the broad peak between 6.5 and 7.25 ppm of the phenylene groups for each polymer, as already reported for PPM obtained using SnCl_4_ or W(II)-based catalysts [21,24].

The obtained polymers were investigated with differential scanning calorimetry (DSC) and thermogravimetric analysis (TGA). All the polymers showed high thermal stability presenting an onset of decomposition temperatures above 400 °C. Moreover, glass transition temperatures in the range of 58 °C–63 °C were found and no further first-order thermal transitions were detected. Thus, the thermal properties of the obtained polymers are in agreement with the data reported for PPM obtained by other catalysts such as SnCl_4_ or W(II)-based complexes (*T*_g_ 60 °C–65 °C, onset of decomposition above 400 °C, Figure 7a,b, respectively) [21,24].

### 3.2. Copolymerization of Benzyl Chloride with BCMD

Based on the results obtained with the above catalysts, [WCl_4_(THF)_2_] was selected in order to enhance the molar mass with the bifunctional branching agent BCMD as the comonomer, since [WCl_4_(THF)_2_] provided the highest *M*_w_. As evident from Figure 8a, the presence of BCMD (0.5% mol/mol) did not affect the initiation temperature of the polymerization (no reaction below 150 °C) but as the temperature of 150 °C was reached, the reaction became faster than in the homopolymerization. While in case of the homopolymerization, the monomer conversion did not grow above 10% at 150 °C, at the same temperature, the monomer conversion in the presence of BCMD reached 70%, accompanied by an increase in viscosity reflected in the observation of the Weissenberg effect. Therefore, the temperature was raised to 180 °C until complete monomer conversion was achieved. The chromatograms displayed in Figure 8b again show a multimodal distribution of the molar masses emerging over the course of polymerization, with a general shift of the peaks to higher molar masses. Notably, differently to the homopolymerization, in the presence of the branching agent, the intensity of the high molar mass peak at 12 min (*M*_n_ 914,800 g mol^−1^) increased with increasing monomer conversion. Although the overall *M*_n_ (3400 g mol^−1^) did not change significantly, *M*_w_ (212,000 g mol^−1^) increased by a factor of four with respect to the homopolymerization. 

The product with BCMD was isolated by dissolution and subsequent precipitation as in the case of the corresponding homopolymer described above. Compared to the in situ product, the isolated product possessed essentially the same *M*_n_ (3400 g mol^−1^) but a slightly lower *M*_w_ (187,900 g mol^−1^). On the other hand, the *M*_w_ was 3.8 times higher than that of the product without BCMD, revealing a higher fraction of high molar mass product obtained with BCMD. In order to separate the highest molar mass fractions, the obtained polymer was dissolved in 2-butanone resulting in a spontaneous separation corresponding to a lower molar mass fraction (F_low_) and a higher molar mass fraction (F_medium_), as reported elsewhere for the fractionation of PPM [21,24]. Moreover, further extraction of lower mass polymers in the F_medium_ fraction was performed with a chloroform:2-butanone 1:1 (by volume) mixture to yield the fraction F_high_ The results of fractionation are shown in Table 2. The *M*_n_ of F_medium_ (33,520 g mol^−1^) is an order of magnitude above the value before fractionation and also above the values commonly obtained for PPM (see Section 1, Introduction). The *M*_n_ of the fraction F_high_ (205,300 g mol^−1^) even exceeds the highest molar mass of a PPM isolated so far (167,900 g mol^−1^, also obtained by fractionation) [22]. The GPC diagram (Figure 9) further reveals that the lowest molar masses (F_low_) were completely separated from the fraction with the highest molar mass (F_high_). It is also evident from Figure 9 that F_high_ consists of a bimodal molar mass distribution with two peaks representing molar masses of 872,000 g mol^−1^ and 122,200 g mol^−1^ (12 min and 13.5 min retention times in the GPC diagram). However, our attempts to separate these two fractions to obtain a fraction with ultrahigh molar mass failed. 

The main difference between the products obtained with and without BCMD in the ^13^C NMR spectra was the rise of a peak at 16.5 ppm (Figure 10a) corresponding to the signal of methyl groups belonging to the durene unit, confirming that durene units were incorporated in the polymer chains. The ^13^C NMR spectra in the methylene region (30 ppm–42 ppm) showed slight differences in the substitution patterns between PPM with durene (PPM-D) and the homopolymer (Figure 10b). Essentially, the relative intensity of the *ortho-ortho* substitution pattern increased somewhat. In the ^1^H NMR spectra, the distinctive signals of PPM emerged (broad peak at 3.7 ppm and in the range of 6.8–7.2 ppm) and a peak at 2.5 ppm which is attributed to the methyl groups in the durene framework. The integration of the NMR signals (Appendix A) revealed a concentration of durene units of about 0.4% mol/mol with respect to phenylene units, which is slightly lower than in the initial reaction mixture. This could originate in the context of the work-up procedure of the sample.

The UV-vis absorption spectra of PPM-D and PPM (Figure 11) show similarities, such as the characteristic small peak at around 450 nm and the four peaks between 320 nm and 420 nm, although the resolution is lower in the spectrum with PPM-D. PPM-D is fluorescent. Two species of PPM, denominated as blue phase and green phase [9], were evident in the photoluminescence (PL) and photoluminescence emission (PLE) spectra (Figure 12). Upon excitation at 366 nm, emission peaks in the PL spectra appeared at 424 nm and 447 nm for the blue phase and, upon excitation at 451 nm, peaks at 477 nm and 505 nm were detected for the green phase (Figure 12a,b), as already reported for PPM synthesized with the catalyst SnCl_4_ (Appendix A) [9]. The PLE spectra of the PPM synthesized with the catalyst [WCl_4_(THF)_2_] (Figure 12c,d) at the detection of 444 nm (predominantly blue phase) did not show marked differences to previously reported PPM [9], and the spectrum of PPM-D showed peaks at similar wavelengths yet with lower resolution. Detection at 504 nm (overlapping blue and green phase) showed a depressed absorption after 445 nm of PPM-D with respect to the homopolymer obtained with [WCl_4_(THF)_2_] or also with SnCl_4_ (Appendix A) [9]. The differences in PLE spectra might be associated with the slight differences in the substitution pattern (see above).

The thermal properties of PPM-D (T_g_ 65 °C, onset of decomposition at 410 °C, Figure 13a,b, respectively) were similar to those of PPM.

Molar masses can also be enhanced by the use of BCMD with common Lewis acids as catalysts like Bi(OSO_3_CF_3_)_3_ (Table 3; for more details, see Appendix A). Thus, *M*_w_ increased by a factor of three (to 35,440 g mol^−1^) compared to PPM synthesized with the same catalyst under the same conditions. This is a substantial increase, although not as pronounced as that with [WCl_4_(THF)_2_]. In any case, this example demonstrates that the strategy of molar mass enhancement by BCMD is not limited to a particular class of chemical compounds which are active as catalysts for the synthesis of PPM.

## 4. Conclusions

The coordination compounds [WCl_4_(MeCN)_2_], [WCl_4_(THF)_2_], [WBr_2_(CO)_3_(dme)] and [MoI_2_(CO)_3_(MeCN)_2_] were employed as catalysts for the polymerization of benzyl chloride. The *M*_n_ values of the poly(phenylene methylene)s were in the range of 2700 g mol^−1^ and 4500 g mol^−1^, i.e., in the common range of PPM, and without characteristic differences attributable to the oxidation state of tungsten (II or IV) or to the type of the metal (W or Mo). However, [MoI_2_(CO)_3_(MeCN)_2_] was found to be particularly active already at 80 °C (monomer conversion above 90%), while the PDI of the polymer obtained with [WCl_4_(THF)_2_] (15.7) was substantially higher than that with the other three catalysts (PDI 2.2–3.9). Moreover, PPM obtained with [WCl_4_(THF)_2_] as a catalyst exhibited a bimodal molar mass distribution with a fraction of high molar masses.

The polymerization of benzyl chloride with a small quantity of the bifunctional monomer BCMD (0.5% mol/mol) using the catalyst [WCl_4_(THF)_2_] yielded a product with *M*_n_ of 3400 g mol^−1^, i.e., in the common range of PPM, however, with a fraction of very high molar mass as reflected in the PDI of 55.3. Accordingly, after fractionation, a product with *M*_n_ of 205,300 g mol^−1^ (PDI = 3.7) was obtained, which is, to our knowledge, the highest *M*_n_ reported so far for a poly(phenylene methylene) product (previously *M*_n_ = 167,900, PDI = 6.4 [24], i.e., with broader molar mass distribution than the PPM polymer with durene units). Notably, using the completely aromatic substituted BCMD as a branching agent, the entire batch was still soluble in organic solvents in contrast to other strategies reported in the literature, in which insoluble and thus non-processable polymers were obtained [25,26,27]. As a side note, all polymers were photoluminescent and showed thermal properties (onset of decomposition above 400 °C, glass transition temperatures from 58 °C–63 °C) as inherent for PPM.

## Figures and Tables

**Figure 1 polymers-16-00967-f001:**
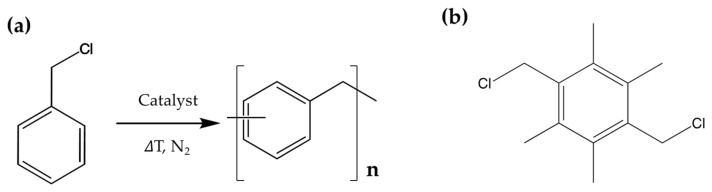
(**a**) Benzyl chloride polymerization scheme and chemical structure of poly(phenylene methylene). (**b**) Chemical structure of 1,4-bis(chloromethyl)-2,3,5,6-tetramethylbenzene (3,6-bis(chloromethyl)durene, BCMD) used as a bifunctional branching agent in this work.

**Figure 2 polymers-16-00967-f002:**
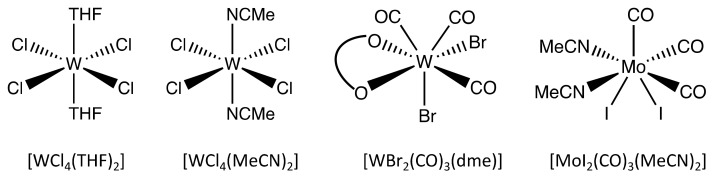
Chemical structures of the used catalysts. THF = tetrahydrofuran, MeCN = acetonitrile, dme = dimethoxyethane.

**Figure 4 polymers-16-00967-f004:**
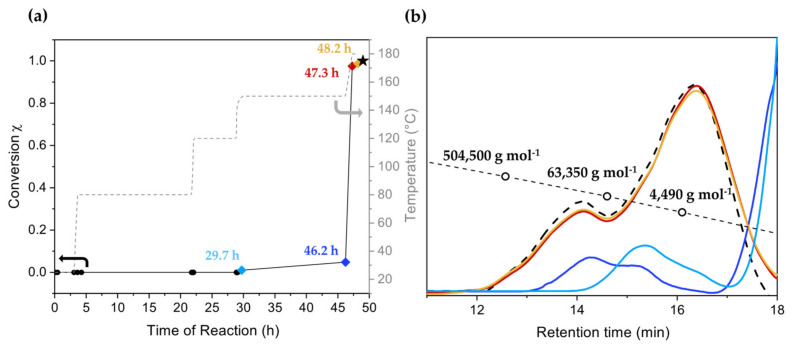
(**a**) Progress of monomer conversion (χ) and operative temperature as a function of the reaction time with the catalyst [WCl_4_(THF)_2_]. The rhombus shapes indicate the sampled aliquots to evaluate the progress of the reaction. The star indicates the product after purification. The arrow helps the reader linking the temperature ramp to the temperature axis. (**b**) Gel permeation chromatograms of aliquots taken during the reaction, revealing the evolution of the molar masses and molar mass distributions according to the reaction times designated in (**a**). The color of each chromatogram corresponds to the color in the reaction diagram (**a**) and indicates the sampling time. The black dashed line indicates the chromatogram of the purified product. The circles provide a qualitative indication of the molar mass range of the calibration line.

**Figure 5 polymers-16-00967-f005:**
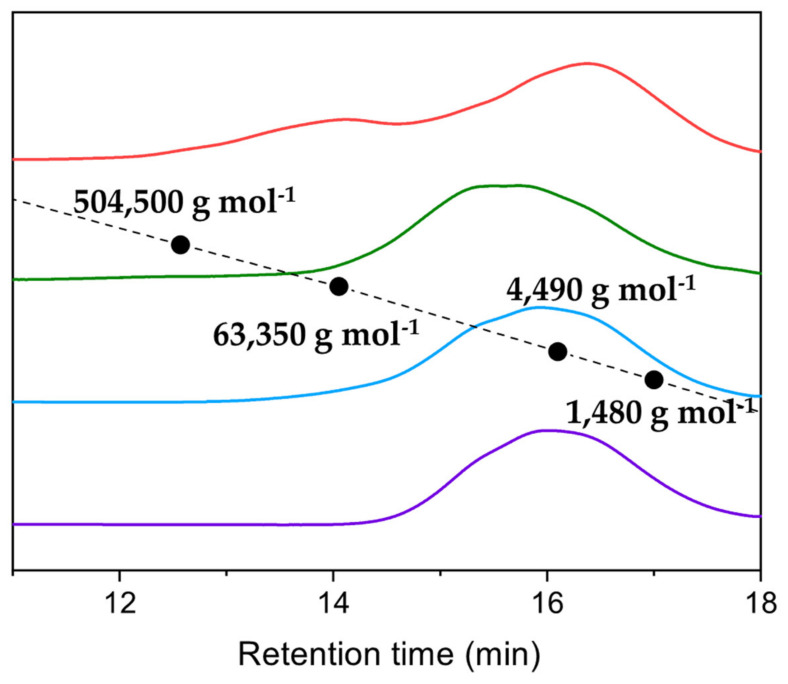
Gel permeation chromatograms of isolated polymers obtained with [WCl_4_(THF)_2_] (red), [WC_l4_(MeCN)_2_] (violet), [WBr_2_(CO)_3_(dme)] (light blue) and [Mo_2_(CO)_3_(MeCN)_2_] (green) as catalysts.

**Figure 6 polymers-16-00967-f006:**
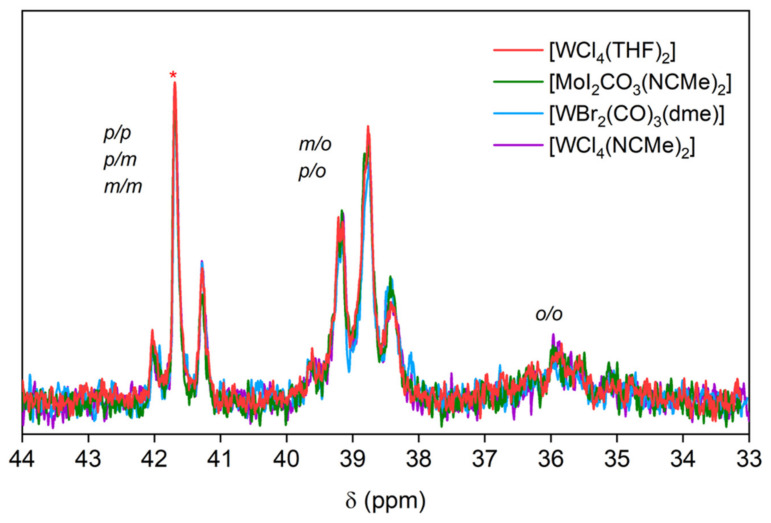
C NMR spectra in the region of the methylene group of PPM synthesized with each catalyst. The peaks are assigned to the respective substitution pattern of the phenylene rings [9,22], with o designating the ortho-, m the meta- and p the para-substituted position.

**Figure 7 polymers-16-00967-f007:**
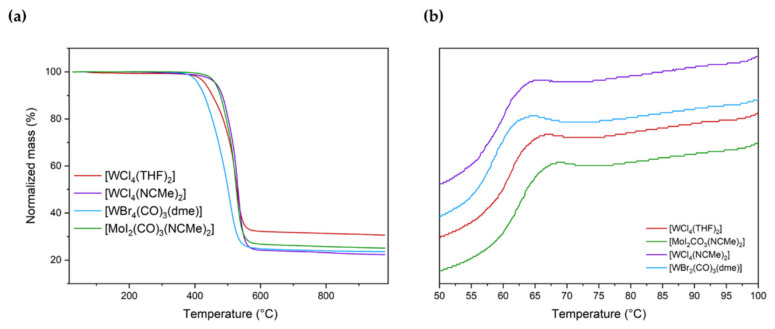
Thermogravimetric analysis (**a**) and differential scanning calorimetry (**b**) results of PPM synthesized with each catalyst.

**Figure 8 polymers-16-00967-f008:**
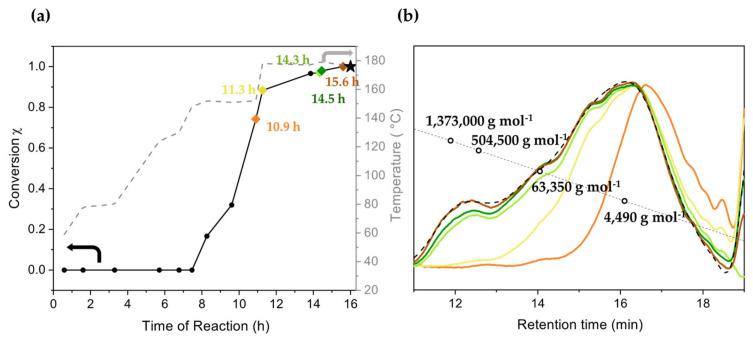
(**a**) Progress of the monomer conversion (χ) and synthesis temperature as a function of the reaction time for the polymerization of benzyl chloride with 0.5% mol/mol BCMD catalyzed by [WCl_4_(THF)_2_]. The arrow helps the reader to link the temperature ramp to the temperature axis. (**b**) Gel permeation chromatograms of aliquots taken during the reaction; the colors of the curve coincide with the colored dots of monomer conversion in (**a**) and thus represent the related reaction times in (**a**).

**Figure 9 polymers-16-00967-f009:**
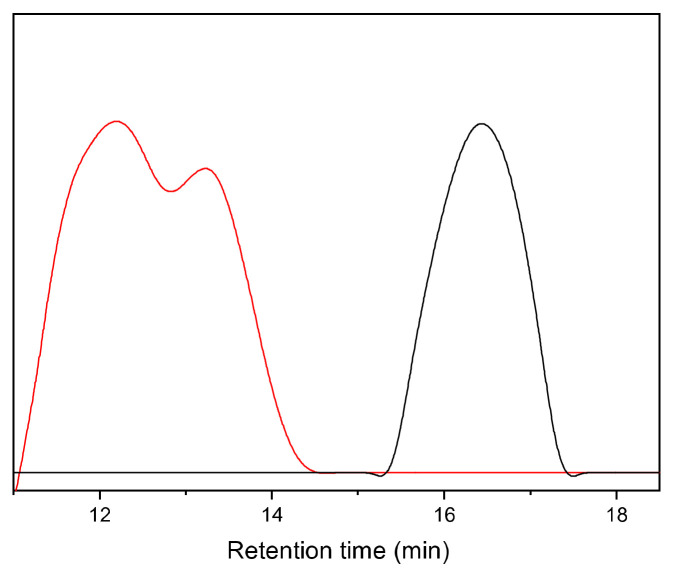
Gel permeation chromatograms of the low molar mass fraction (F_low_, black) and high molar mass fraction (F_high_, red) of the product obtained by the polymerization of benzyl chloride with 0.5% mol/mol BCMD catalyzed by [WCl_4_(THF)_2_].

**Figure 10 polymers-16-00967-f010:**
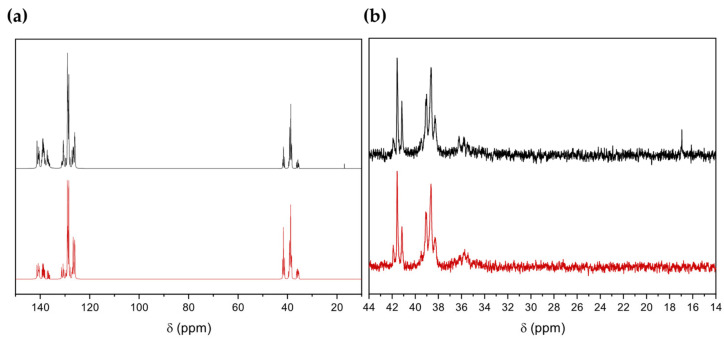
^13^C NMR spectra of PPM-D (black) and PPM (red). (**a**) Region of the aromatic units; (**b**) region of the methylene groups.

**Figure 11 polymers-16-00967-f011:**
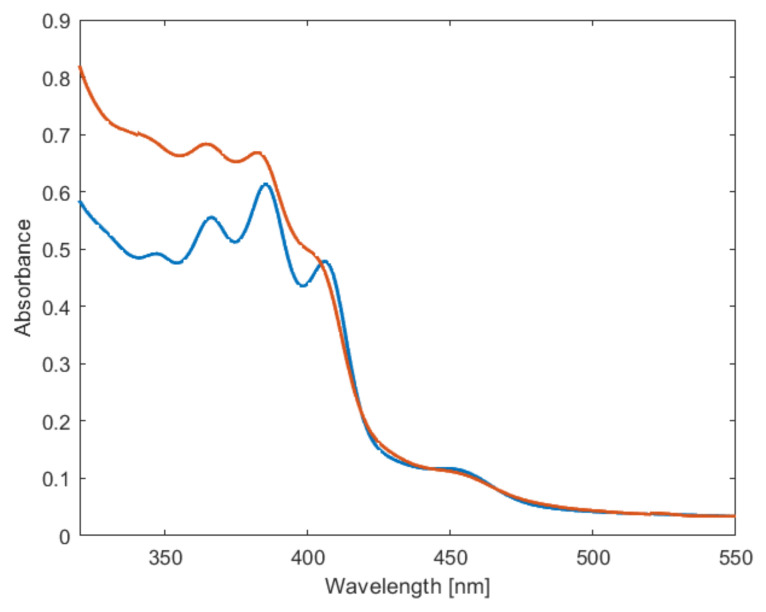
UV-vis absorption spectra of PPM (blue) and PPM-D (orange) in chloroform (0.5% m/m).

**Figure 12 polymers-16-00967-f012:**
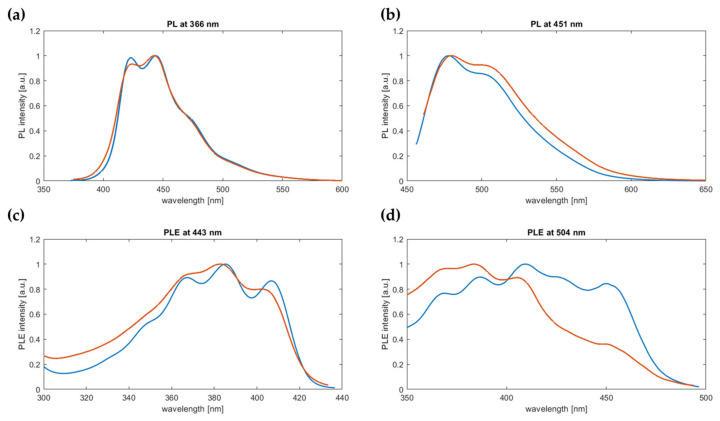
Normalized optical spectra of PPM (blue) and PPM-D (orange), both synthesized with the catalyst [WCl_4_(THF)_2_]. Above, PL recorded upon exciting the polymer at 366 nm (**a**) (corresponding to the total photoluminescence emission), and at 451 nm (**b**) (corresponding to the green phase emission). Below, PLE spectra for emission detected at 443 nm (**c**) and 504 nm (**d**).

**Figure 13 polymers-16-00967-f013:**
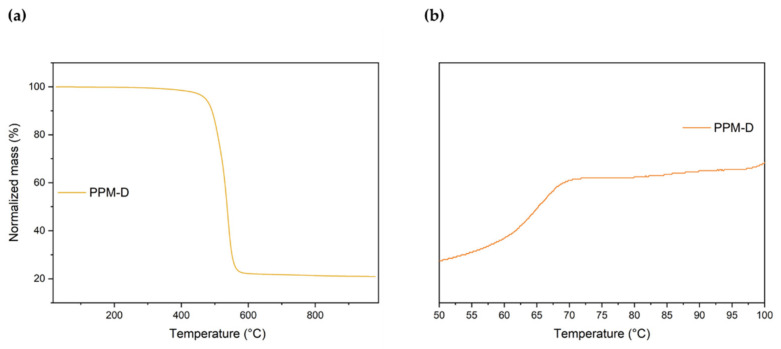
Thermogravimetric analysis (**a**) and differential scanning calorimetry (**b**) results of PPM synthesized in the presence of BCDM.

**Table 2 polymers-16-00967-t002:** Number average molar mass (*M*_n_), weight average molar mass (*M*_w_), polydispersity index (PDI), yield of fractionation and polymer fraction composition resulting from the polymerization of benzyl chloride with 0.5% mol/mol BCMD catalyzed by [WCl_4_(THF)_2_], yielding the fraction F_low_, F_medium_, F_high_ (see text).

Fraction	*M*_n_ (g mol^−1^)	*M*_w_ (g mol^−1^)	PDI	Yield of First Fractionation (%)	Yield of Second Fractionation (%)
F_low_	1821	3794	2.1	55	
F_medium_	33,520	322,100	9.6	45	54
F_high_	205,300	772,900	3.7		46

**Table 3 polymers-16-00967-t003:** Number average molar mass (*M*_n_), weight average molar mass (*M*_w_) and polydispersity index (PDI) resulting from the polymerization of benzyl chloride with 0.5% mol/mol BCMD (PPM-D) and of PPM catalyzed by Bi(OSO_3_CF_3_)_3_.

Product	*M*_n_ (g mol^−1^)	*M*_w_ (g mol^−1^)	PDI
PPM	3244	12,100	3.7
PPM-D	3772	35,440	9.4

## Data Availability

Data are contained within the article.

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
