# Peer review of "Synthesis of Soluble High Molar Mass Poly(Phenylene Methylene)-Based Polymers"

_polymers, 2024, doi:10.3390/polym16070967_

Round 1

Reviewer 1 Report

Comments and Suggestions for Authors

Manuscript ID: polymers-2931622

Title: Synthesis of soluble high molar mass poly(phenylene methylene)-based polymers

This paper reports the detailed information on the synthesis of soluble poly(phenylene methylene) (PPM)-based multifunctional polymers with high molecular mass using the . The obtained compounds expand a range of its potential applications (organic coatings for corrosion prevention, adhesives, etc.). The characterization of the synthesized polymers is performed using various analytical methods (NMR, GPC, DSC, and TGA). Undoubtedly, this paper can be interesting to the readers of this journal. However, it should be improved prior to publication. Some comments are given below.

1) It is highly recommended to add the DSC and TGA curves for all the synthesized polymers to the main text of the manuscript for better presentation of the numerical results.

2) In PDF-file of the manuscript and DOC-file of the Supplementary Materials, Figures 3, 6, S1 and S2 have black background instead of the axis signatures. The authors should clarify (or correct) this moment (it might have seemed a little strange).

Author Response

I would like to kindly thank the reviewer for his interesting insights. In fact, we agree with the reviewer that including TGA and DSC graphs can help the reader to a better understanding of the topic. We would also take this opportunity to thank him for the suggestion about the black background graphs. It must have been an error in the uploading of the material from MAC. we have taken care to change the reported graphs.

Reviewer 2 Report

Comments and Suggestions for Authors

Thiis paper is interesting and I enjoyed reading it.  In future work, it might be interesting to examine the outcome of using slow addition of the monomers to the feed rather than charging the reaction.  Since NMR showed that BCMD incorporation was at a lesser extent than the feed, I wonder if adding benzyl chloride slowly to the reaction might have some benefit.  

Author Response

I would like to kindly thank the reviewer for his interesting insights. We will certainly welcome his suggestions for our future outlooks.